# Boosting test-efficiency by pooled testing for SARS-CoV-2—Formula for optimal pool size

**Rudolf Hanel** [1,2☯], **Stefan Thurner** [1,2,3,4☯] *

**1** Section for Science of Complex Systems, Medical University of Vienna, Vienna, Austria, **2** Complexity Science Hub Vienna, Vienna, Austria, **3** Santa Fe Institute, Santa Fe, NM, United States of America, **4** IIASA, Laxenburg, Austria

☯ These authors contributed equally to this work.
\* stefan.thurner@meduniwien.ac.at

**Data Availability Statement:** The study analyses a mathematical relation between infection levels in a population and optimal pool-sizes in a class of group testing strategies. Information about expected infection levels and other of various populations in the current COVID-19 crisis, needed

## Abstract

In the current COVID19 crisis many national healthcare systems are confronted with an acute shortage of tests for confirming SARS-CoV-2 infections. For low overall infection levels in the population the pooling of samples can drastically amplify the testing capacity. Here we present a formula to estimate the optimal group-size for pooling, the efficiency gain (tested persons per test), and the expected upper bound of missed infections in pooled testing, all as a function of the population-wide infection levels and the false negative/positive rates of the currently used PCR tests. Assuming an infection level of 0.1% and a false negative rate of 2%, the optimal pool-size is about 34, and an efficiency gain of about 15 tested persons per test is possible. For an infection level of 1% the optimal pool-size is 11, the efficiency gain is 5.1 tested persons per test. For an infection level of 10% the optimal pool-size reduces to about 4, the efficiency gain is about 1.7 tested persons per test. For infection levels of 30% and higher there is no more benefit from pooling. To see to what extent replicates of the pooled tests improve the estimate of the maximal number of missed infections, we present results for 1 to 5 replicates.

## Introduction

We briefly analyse how *pooled testing* increases the efficiency of testing for viral infections, given that only a limited number of tests is available. The idea is to pool samples taken from several subjects and test the combined sample with a single test. If the test is negative all subjects are negative. If the test is positive all individuals are tested to find the infected ones. Pooling in the context of COVID19 was recently suggested by Dina Berenbaum of the Technion Israel Institute of Technology [1] and has been implemented at the Rambam Medical Center and the Technion in Haifa, who have been inviting other hospitals to follow their example [2]. Initial tests there indicate that pooling works for the *severe acute respiratory syndrome coronavirus 2 (SARS-CoV-2), using available essays*. It was initially suggested that up to 32—maybe over-optimistically even 64—people could be tested with a single test.

Pooled testing, also called group testing, was first introduced by Dorfman (1943) to screen U.S. soldiers for syphilis [3], and has become a well studied field using a spectrum of

for applying our findings in order to optimise group-sizes for group testing, need to be acquired on a case-to-case basis. Other pertinent information to the paper is either referenced in the bibliography or included in the supporting information, SI, where we have included (a) a manuscript that, besides discussing some subtleties of pooled testing, also provides a Matlabfunction in appropriate text form. Furthermore, (b) also the four Matlab scripts for computing the figures in the main manuscript and the SI manuscript, and also the code provided in the SI manuscript as a convenient Matlab function file.

**Funding:** ST 857136 Austrian Research Promotion Agency, FFG https://www.ffg.at/en The funders had no role in study design, data collection and analysis, decision to publish, or preparation of the manuscript.

**Competing interests:** The authors have declared that no competing interests exist.

techniques. Those methods have been widely applied in various fields, for example, in aspects of DNA sequencing, including applications in clone library screening or protein-protein interaction detection [4]. They were also used in tasks of screening human populations and saving potential screening costs. For example, HIV screening in the United Stated and in Thailand [5]. Group testing techniques include linear, combinatorial, or adaptive methods. They have been based on information theoretic considerations [6], or array based methodology [7, 8], and can be used to address a number of mathematical problems concerning various risk characteristics, partitioning problems, and the trade-off between the test accuracy and efficiency [9–11]. In the context of the recent crisis potentially new approaches such as *noisy group testing algorithms* [12] or *double pooling tests* [13] are being explored. A review of algorithms can be found in [14].

While the observation [1, 2] that pooling appears to be a viability strategy for SARS-CoV-2, a number of factors have been reported that may limit pool-sizes, so that they can be used in PCR-based tests in a meaningful way. This includes issues concerning sample collection and assay limitations [15–19], issues of sample dilution, and disease prevalence or incidence (the expected fraction of positive individuals to be tested, i.e. the *infection level*), and assay type, such as S-gene and/or E-gene based essays. The differential sensitivity and specificity between individual tests and pooled tests of RNA RT-PCR testing protocols, available for SARS-CoV-2, may also depend on, and therefore limit, pool-sizes that realistically can be used [20, 21]. For positive test rates between 4 and 24%, for instance, positives in pool sizes up to 30 individuals could be detected but *cycle threshold values*, i.e. the number of amplification cycles necessary to reach detection threshold, may differ up to 5 amplification cycles, [21]. In particular, the false-negative rate of pooled tests may depend on the pool-size and viral load of patients or samples, while false positives seem to be essentially unaffected [22]. This suggests to use smaller pool sizes than the suggested 32 or 64. Repeated testing (replicates) for controlling false negative rates of pooled tests has been discussed, see, e.g. [23].

Here we contribute an estimation of the benefits of a simple, easy to implement, one-stage pooling strategy. Our main goal is to compute the optimal group-size for pooled tests and their dependence on the disease prevalence, i.e. the fraction of infected in the target population. We provide a formula for the optimal group-size, i.e. the optimal number of persons pooled into a single test, and study the dependence of this optimum on false positive and false negative rates of the used PCR test. We demonstrate the optimal group-size dependence on the false negative rate of pooled testing. We briefly remark on a subtlety when using replicate measurements in group testing to control for the false negative rate of tests. We ask how the number of test replicates affects the false negative rate of the pooled test. We conclude that the optimal group size, in the considered single-stage pooling approach, should be smaller than the suggested 32-64, for the currently (April 2020) suspected infection levels in general populations. Finally, we comment that testing a pooled sample more than twice will essentially not further decrease the expected maximal number of possibly missed positive cases.

## Material and methods

### Model

We assume that

- a fraction λ of infected people in a population,

- tests have a *false positive* rate of $\gamma_+$ and a *false negative* rate of $\gamma_-$. If not stated otherwise, we assume that testing a pooled sample does not change the false positive and false negative

rates of the test. We discuss the effect of group-size dependent false negative rates at the end of the paper.

- We pool samples into groups of size, $\omega$.

- To control false negatives we take $r$ replicates of the pooled test. We then apply a majority rule, meaning that if the majority of the $r$ replicates are positive, the pooled sample is declared positive. We will speak of a majority either, (m1) if the number of positive tests is greater than the number of negative tests or, (m2) if the number of positive tests is at least as large as the number of negative tests. The two versions of majority rule show a considerable differences for even numbers of replicates. Although optimal group-size behaves very similar for both majority rules, rule m2 turns out to be consistently superior to m1 in terms of false negatives and therefore will be considered in detail in the discussion below.

- If the pooled sample is declared positive, each individual in the respective group is tested separately.

    Under these assumptions we compute

- the optimal group size, $\omega^{\text{opt}}$,

- the effective number of persons that can be tested with one test, *PPT* (persons per test),

- an estimate for the upper bound for the fraction of infected individuals that are missed by the pooled testing procedure (applied to the population). We call it the *pooled testing risk factor* and denote it by *PTRF*. We also compute the *false negative rate*, *FNR*, of pooled testing, which is the fraction of infected individuals the pooled test will miss on average,

- and finally, we discuss and demonstrate how false negative rates increasing with group size affect the optimal group-size, false negative rates (FNR), and the pooled testing risk factor (PTRF).

## Optimal group size

We call a group *positive* if at least one of its members is positive. The probability of a group to be positive is

$$p = 1 - (1 - \lambda)^{\omega}. \tag{1}$$

Because of limitations in test sensitivity and specificity, tests will be falsely declared positive in $(1 - p)\gamma_+$ cases. False positives do not decrease the chances to capture a true positive but only decrease the efficiency in using the available tests. More importantly, tests will miss positive individuals in $p\gamma_-$ cases, on average. Note that $\gamma_+$ and $\gamma_-$ might need to be considered carefully with respect to how tests are performed (essay type) and who gets tested (patients with high or low expected viral load).

    To see how test replicates affect FNR and PTRF of the pooled test we test a sample $r$ times and then apply the majority rule (m2). For this we have to compute the conditional probabilities

$$\gamma_+^*(r) = \sum_{i \geq r/2} \binom{r}{i} \gamma_+^i (1 - \gamma_+)^{r-i}, \tag{2}$$

that the majority rule declares the replicate test falsely as positive, and

$$\gamma_-^*(r) = \sum_{i>r/2} \binom{r}{i} \gamma_-^i (1-\gamma_-)^{r-i} , \tag{3}$$

that the majority rule declares the replicate test falsely as negative. Note that these equations hold under the majority assumption (m2) that we have at least as many positive than negative tests. For majority rule (m1) that we have more positive than negative tests one has to sum over $i > r/2$ instead of $i \geq r/2$ for $\gamma_+^*(r)$, and vice versa for $\gamma_-^*(r)$, on the left side of the equations. Also note that for $r = 1$, we have $\gamma_\pm^*(r=1) = \gamma_\pm$. Hence, the probability, $P_+^*(r)$, that a test with $r$ replicates registers as *positive* is given by

$$P_+^* = p(1-\gamma_-^*) + (1-p)\gamma_+^* = p(1-\gamma_+^* - \gamma_-^*) + \gamma_+^* , \tag{4}$$

and $P_+ = P_+^*(r=1)$ is the probability for a single test to register as positive.

The expected number of tests per person therefore can be estimated by the upper bound, $Q$, given by

$$Q = \frac{1}{\omega}\left(r(1-P_+^*) + (r+\omega)P_+^*\right) = P_+^* + \frac{r}{\omega} , \tag{5}$$

and the number of *persons per test* (PPT) then simply follows the lower bound,

$$PPT = 1/Q = \frac{\omega}{r + P_+^*\omega} . \tag{6}$$

For $r = 1$ Eq (5) is exact. The reason why, for $r > 1$, Eq (5) is in fact an upper bound for the expected number of tests per person (and PPT a lower bound for the expected number of persons processed per test) is the following. One could save some of the $r$ replicates in the following way. Consider $r$ to be an odd integer, so that the majority rules m1 and m2 are identical. For certain sequences of positive or negative results, the majority rule is fulfilled or can no longer be fulfilled before all $r$ test replicates have been performed. For instance, for $r = 3$ one could skip the third test if the first two tests already were positive. Why? Since one already has achieved a majority of positive tests. Similarly, if the first two tests are negative, the third can be skipped. This would effectively reduce $r = 3$ to $r^* = 2\frac{4}{8} + 3\frac{4}{8} = 5/2 = 2.5$ (note that $r/r^* =$ 1.2), and $r = 5$ to $r^* = 3\frac{8}{32} + 4\frac{4}{32} + 5\frac{20}{32} = 35/8 = 4.375$ (note that $r/r^* \sim 1.143$). However, since in practice the number of replicates will typically be $r = 1$ ($Q$ and PPT are exact and $r^* = r = 1$) or $r = 2$ and the bounds for all $r > 1$ are more favourable than the respective bounds— and do not excessively deviate from the true expectation values. Here we do not consider the complication of path-dependent replicate numbers and consider $r$ tests performed indiscriminately. An on-line pool-size calculator (for the case $r = 1$) working on the basis of this analysis can be found online [24].

Similarly, one can compute an upper bound for the expected number of cases that we might miss when testing pooled samples, *PTRF*. It is expressed as the expected number of missed infections per tested person (not per tested infected person). If we assume that a group is positive and we test it, then we miss it when the majority rule gives us a negative, which happens, when in the majority of cases we get a false negative. We therefore get that the pooled testing risk factor is given by

$$PTRF \equiv p(\gamma_-^* + (1-\gamma_-^*)\gamma_-) . \tag{7}$$

Note that the expected *maximal* number of missed infections, *PTRF*, must not be confused with the *false negative rate* of the pooled test, $FNR = \gamma_-^* + (1 - \gamma_-^*)\gamma_- \sim 2\gamma_-$, which is the *average* number of individuals one expects to miss in pooled testing per infected person. Note that the approximation holds for $r = 1$ only and small $\gamma_-$. If there are no biases or correlations within or between groups, we get that the number of missed infections will be $\lambda\, FNR \sim 2\gamma_-\lambda$, which does not depend on group-size. It can be checked that $PTRF = p\, FNR$. The advantage of *PTRF* over *FNR* is that in testing of *biased groups* one can be confronted with correlated cases with an increased chance of multiple infections within a group, w.r.t. the entire population. *PTRF* therefore captures this situation by considering the *upper bound* of missed positives rather than the average.

## Results

Results for the optimal pool size, $\omega^{\text{opt}}$, and the persons per test, *PPT*, see Fig 1. In Fig 1A the optimal pool size is shown for a population-wide infection level of 1%. In Fig 1B the increase of *PTRF* with pooling size is seen. Here we use a false negative rate of $\gamma_- = 0.02$ and a false positive rate of $\gamma_+ = 0.0012$, which are sensible estimates for PCR tests that are currently used in Austria (as of March 20) [25]. We show the case for $r = 1$, 3, and 5 replicates for the pooled test in full lines, blue, orange, and green. Replicates for $r = 2$ and 4 (even) are shown for majority rule m2 with dotted lines in red and purple, respectively. We find that $r = 1$ assumes $\omega^{\text{opt}} = 11$ where it achieves a *PPT* of approximately 5.14 persons per test and a *PTRF* of 4.14 $10^{-3}$. For $r = 2$ under m2 one gets $\omega^{\text{opt}} = 15$, *PPT* is 3.63, and *PTRF* 2.13 $10^{-3}$, whereas for $r = 2$ under m1 (which is not shown in the figure) one gets that $\omega^{\text{opt}} = 16$, the *PPT* is 3.78, and FNPT 1.38 $10^{-2}$. Note that *PTRF* for $r = 2$ and m1 is worse than *PTRF* for $r = 1$. In general, $\omega^{\text{opt}}$ and *PPT* are very similar for majority rules m1 and m2 even for even $r$. However, the *PTRF* values of even replicates $r$ for m2 are consistently lower than those for m1. In the case $r = 2$ by a factor of approximately 2.67. Typically, the *PTRF* values for even $r$ under m1 are even higher than the *PTRF* of the next odd replicate test with $r - 1$. For this reason we suggest to not use majority rule m1. Note that the false negative rate, FNR, does not depend on group-size with values

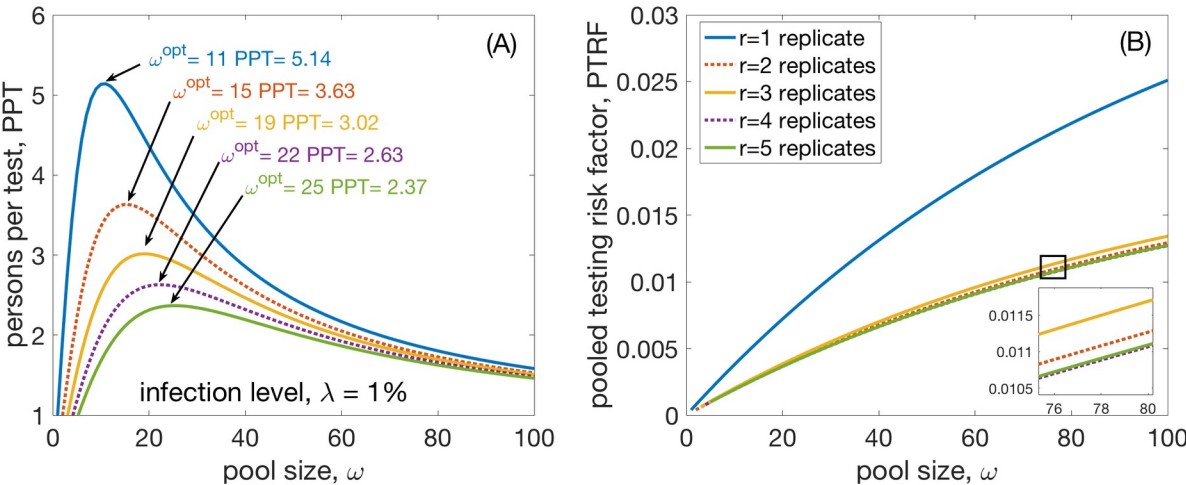

**Fig 1. Group test efficiency.** (A) Increase of test efficiency in persons per test, *PPT*. The maximum of this curve indicates the optimal pool size, $\omega^{\text{opt}}$ for a given infection level (1%) and given false negative and positive rates of the test. Results are shown for $r = 1$, 2, 3, 4 and 5 replicates of testing the pooled sample. the maximum efficiency gain is naturally found for $r = 1$ and is about 5.1 persons per test, followed by $r = 2$ with a gain of approximately 3.6. (B) The pooled test risk factor for the pooled sample, *PTRF*. The result shows that taking more replicates decreases the false negatives. However, note that this also decreases the efficiency in terms of *PPT*. $\gamma_+ = 0.0012$ and $\gamma_- = 0.02$.

3.96% (1 replicate), 2.04% (2 replicates), 2.15% (3 replicates), 2.00% (4 replicates), 2.01% (5 replicates). Also for the FNR, we see that a pooled test with $n$ replicates, with $n$ being odd, has a higher FNR than the respective test for $n − 1$ replicates. However, FNR for $n + 2$ replicates is always smaller than for $n$ replicates.

The group-size dependent PTRF, on the other hand, is again decreasing for $r \geq 2$, after an initial decrease from $r = 1$ to $r = 2$. This somewhat counter-intuitive behaviour comes from the the dependence of the optimal pool size $\omega^{\text{opt}}$ on $r$. In fact it can be computed by searching the maximum of $PPT$ and solving for $\omega$, i.e. by solving $\frac{d}{dr} PPT(r) = 0$ (compare SI1 in S1 File). It is approximately given by

$$\omega^{\text{opt}} = \sqrt{\frac{r}{\lambda \ (1 - \gamma_-^*(r) - \gamma_+^*(r))}}, \tag{8}$$

which is increasing faster in $r$ than $FNR = \gamma_-^* + (1 - \gamma_-^*)\gamma_- \sim \gamma_-^* + \gamma_-$ decreases, which explains why $PTRF = p\ FNR$ increases with increasing $r$, for $r \geq 2$ (compare S12 Fig and S12.1 Fig in S1 File).

Fig 2 shows the dependence of optimal group size, persons per test, and $PTRF$ on infection levels. In Fig 2A we see the optimal group size, $\omega^{\text{opt}}$, as a function of the infection level of the population. The inset shows the case for low infection levels between 0 and 3%. The case for $r = 1, 3,$ and 5 replicates is shown in solid lines, blue, orange, and green, while the even replicates $r = 2$ and 4 are shown as dotted lines, red and purple, respectively. Fig 2B shows the optimal efficiency gain of persons per test, $PPT$, also with an inset showing small infection levels. Fig 2C gives the risk factor $PTRF$ for pooled testing. It is clearly visible that, for a fixed infection level, taking more than two replicates, $r > 2$, does not improve $PTRF$. However, $r = 2$ does improve it by approximately a factor of 1.5. The jumps in the curves are due to discrete jumps of the optimal group sizes.

We computed the same values for a false negative rate of $\gamma_- = 0.05$. The results for $\omega^{\text{opt}}$ and $PPT$ practically do not change, however, in this scenario, $PTRF$ approximately doubles or triples for all infection levels.

To get a better understanding of the effect that group-size dependence has on of false negative rates, we compare three scenarios. We assume that false positive rates are constant ($\gamma_+ = 0.0012$) and false negative rates increase linearly with group size. We assume that at the

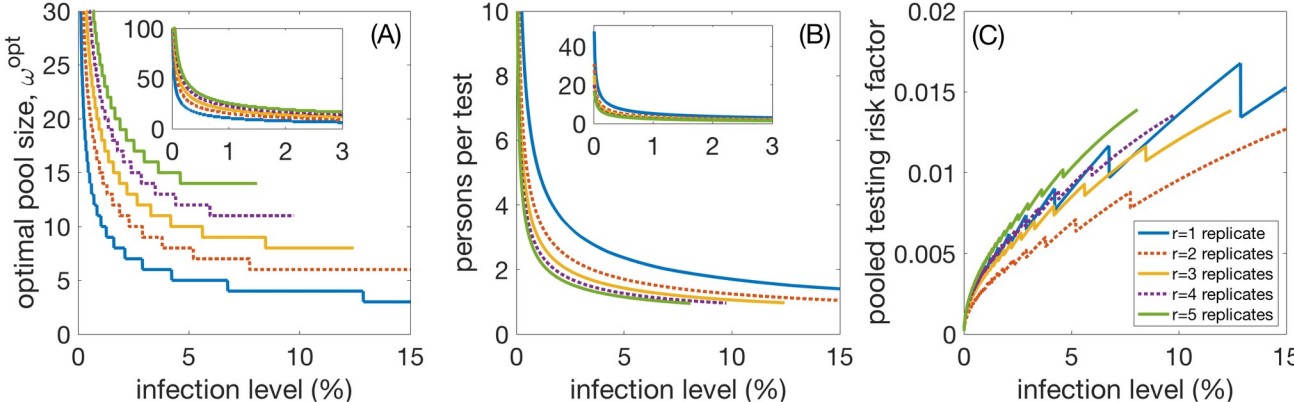

**Fig 2. Infection level dependence.** (A) Optimal pool size, $\omega^{\text{opt}}$, as a function of the infection level of the population. The inset is a blow-up for low infection levels. The cases for $r = 1, 3,$ and 5 replicates is shown in blue, red, and orange, respectively. (B) Efficiency gain of persons per test, $PPT$; the inset shows low infection levels. (C) The pooled testing risk factor $PTRF$. It is clear that taking more replicates does practically not lower $PTRF$, except for $r = 2$. $\gamma_+ = 0.0012$ and $\gamma_- = 0.02$. By taking $\gamma_- = 0.05$, $\omega^{\text{opt}}$ and $PPT$ remain practically unchanged, $FNPT$ doubles for all infection levels in this case (not shown).

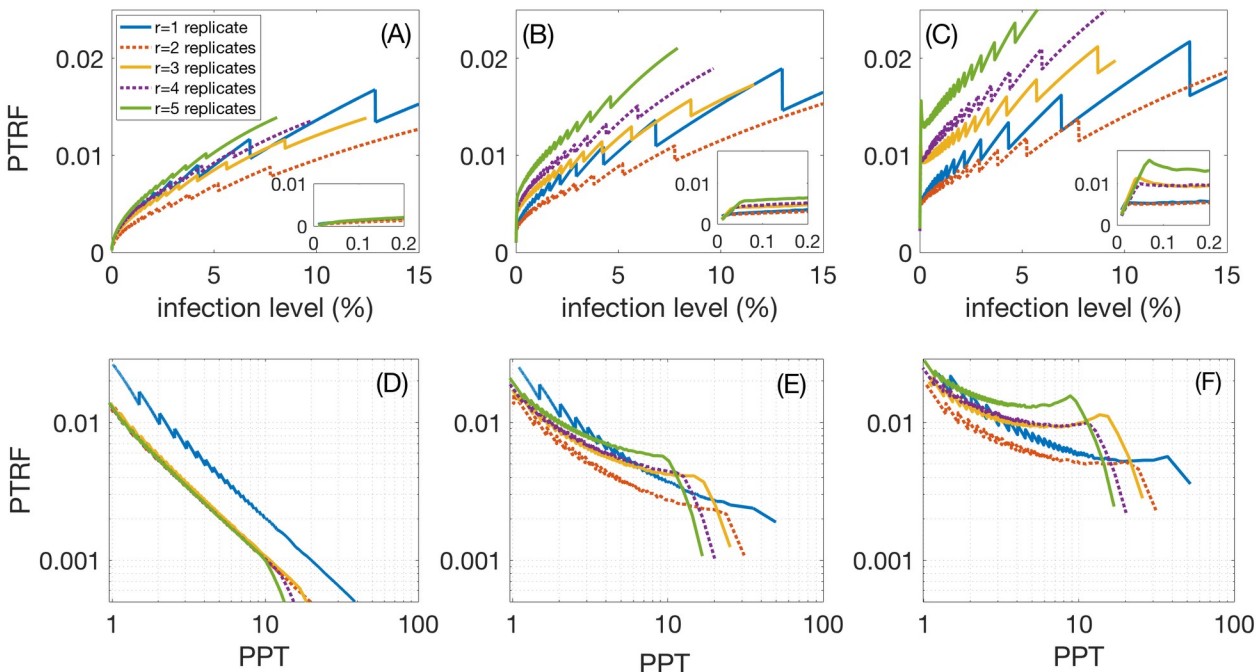

**Fig 3. Group-size dependence of the false negative factor seen in three scenarios of false negative rates that increase linearly with group-size; no group-gize dependence (A,D), a doubling (B,E), and a quadrupling (C,F) of the false negative rate values at group-size 20.** (A), (B), and (C) show that the overall best choice of replicates with respect to *PTRF*, are *r* = 1 and *r* = 2. The insets are blow-ups for low infection levels. Panels (D), (E), and (F) show that this remains true if we consider the optimal *PTRF* at a given gain in persons per test, PPT, except maybe for very low infection levels, corresponding to *PTRF* below 0.5%, i.e. optimal group sizes larger than 20.

maximally considered group size of 100, scenario (1) has the same value as for group size 1 ($\gamma_-$ = 0.02); (2) has 5 times that value, and (3) has 10 times that value. While optimal group-size and PPT do essentially not change, the *PTRF* depends strongly on variable group sizes. This is seen in Fig 3A, 3B and 3C, whose panels correspond to scenarios 1, 2, and 3, respectively. Insets magnify the results for low infection levels. Fig 3D, 3E and 3F, (corresponding again to scenarios 1, 2, and 3) shows that, while *r* = 1 is the most efficient choice, *r* = 2 gives a clear improvement for *PTRF*, and that *r* > 2 does not improve the relation between the gain in terms of PPT and *PTRF*, except for *PTRF* levels below 0.2–0.5%.

## Conclusions

- The optimal pool size and efficiency of pooling strongly depends on the infection level of the population. Let's assume the simplest case of only one test (1 replicate). From Fig 2A and 2B we read off that for an infection level of 0.1%, the optimal pool size is about 34, the efficiency gain is about 15 tested persons per test. For an infection level of 1%, the optimal pool size is 11, the efficiency gain is about 5 fold. For an infection level of 10%, the optimal pool size is reduced to 4, the efficiency gain is a factor of 1.7. For infection levels of 15% this factor drops below 1.5 and the optimal pooling size becomes 3. This size of 3 remains the optimal pooling size up to infection levels of 29% where the efficiency drops to 1.1. From infection levels of 30% and larger pooled testing ceases to be effective.

- Replicates help to lower the pooled testing risk factor, *PTRF*. Using two replicates can significantly lower the pooled testing risk factor. However, increasing numbers of replicates to three is only warranted for group sizes larger than 20. We find that for one replicate at an

infection level of 0.1% we will maximally miss about 1 positive case in 800 pooled tests (0.13%) at most. At 1% we will maximally miss about 1 case in every 241 pooled tests (0.41%); see also Fig 1B.

- When even numbers of replicates are used, majority rule m2 should be implemented, i.e. a pool should be considered positive if at least half of the replicates are positive. For odd numbers of replicates rules m1 and m2 are identical. Rule m1, that there must be more positive than negative replicates, does not essentially change the optimal group-size for pooling. In relative to m2, m1 has a higher pooled testing risk factor, i.e. the maximal number of positive individuals that can be expected to be missed per tested individual. For two replicates, $r = 2$, m1 has a an even higher pooled testing risk factor than using one replicate, $r = 1$.

Let us emphasize that a pooling strategy is most powerful for population-wide screening and mixed samples, for example at airports. Using them for highly biased samples, e.g. for samples from patients already showing symptoms, will be much less effective. Note, that situations with many asymptomatic individuals, with possibly low viral loads, require test protocols that operate in ranges with values of false negative rates larger than 0.02, which would make $r = 2$ replicates the overall favourable choice.

## Example

We finish with a practical example. For Austria, a country with slightly less than 10 million inhabitants an actual infection level of 0.1% would indicate an optimal pool size of 34. For a level of 1% it would be 11. Assuming the true number of infected to be somewhere between 10,000 and 100,000 this would mean a reasonable choice of pooling sizes of about 20. This number is definitely lower than the suggested sizes reported in [1, 2] and about the upper range of pool sizes, that due to possible dilution of viral RNA in the pooled samples, are typically used in screening. The expected gain for this pool size would be about a factor of 10, if 1 replicate is indicated, and a gain of about 5.5 for 2 replicates.

## Supporting information

**S1 File.**
(PDF)

**S2 File.**
(ZIP)

## Acknowledgments

We thank Robert Strassl, Mathias Beiglböck, Walter Schachermayer, and Reinhard Winkler for helpful discussions. We also would like to thank Tim Zander and Dror Baron for their remarks that helped to improve the manuscript.

## Author Contributions

**Conceptualization:** Rudolf Hanel, Stefan Thurner.

**Formal analysis:** Rudolf Hanel.

**Funding acquisition:** Stefan Thurner.

**Methodology:** Rudolf Hanel.

**Project administration:** Stefan Thurner.

**Writing – original draft:** Rudolf Hanel.

**Writing – review & editing:** Stefan Thurner.

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
