## [Decision Letter · Decision Letter 0]

29 Jun 2020

PONE-D-20-13830

Boosting test-efficiency by pooled testing strategies for

SARS-CoV-2

PLOS ONE

Dear Dr. Hanel,

Thank you for submitting your manuscript to PLOS ONE. After careful consideration, we feel that it has merit but does not fully meet PLOS ONE’s publication criteria as it currently stands. Therefore, we invite you to submit a revised version of the manuscript that addresses the points raised during the review process.

Specifically, one of the reviewers raised a number of issues that need to be addressed before the manuscript becomes acceptable. Of particular significance is the inclusion of COVID-19 related data to support the findings.

We look forward to receiving your revised manuscript.

Kind regards,

Oscar Millet

Academic Editor

PLOS ONE

Journal Requirements:

Reviewers' comments:

Reviewer's Responses to Questions

**Comments to the Author**

1. Is the manuscript technically sound, and do the data support the conclusions?

Reviewer #1: Yes

Reviewer #2: Partly

2. Has the statistical analysis been performed appropriately and rigorously? 

Reviewer #1: Yes

Reviewer #2: N/A

3. Have the authors made all data underlying the findings in their manuscript fully available?

Reviewer #1: Yes

Reviewer #2: No

4. Is the manuscript presented in an intelligible fashion and written in standard English?

Reviewer #1: Yes

Reviewer #2: No

5. Review Comments to the Author

Reviewer #1: excellent and important practical manuscript. citation which arrives at similar conclusion and provides direct experimental evidence : Lancet Infect Dis 2020

Published Online

April 28, 2020

https://doi.org/10.1016/

S1473-3099(20)30362-5

might bbe cited

A limitation which might be briefly discussed is the differential sensitivity and specificity of specific "PCR" tests which use one or two) target sequences, and there are small differences between viral target genes and the abundance of target RNA species. Efficiency of the specific methods used for RNA extraction and reverse transcription are variables that may effect sensitivity and ultimately limit pooling.

Reviewer #2: 1. Is it reasonable to assume the false positive/negative rate does not change when switched from the regular testing to pooled testing? Does it apply to COVID-19 screening? It is hard to believe using a pool of size 32 or 64 can have the same false rates as testing the individuals one-by-one.

2. What is the choice of r in practice? How does it affect test efficiency? Following the majority rule, suppose r=5, then in practice, once you observed 3 of them were positive, there is no need to test the remaining 2 replicates. How does this affect the calculation of number of tests needed?

3. Line 47 on Page 2, I believe it should be "If the pooled sample is declared positive, we test each individual in the group separately" because of the majority rule you proposed.

4. The derivation of (3-5) in-explicitly assumed that given the true infection statues, the test results are mutually independent. Is this assumption supported by COVID-19 tests? If COVID-19 tests declare a sample as positive if the measured viral loads exceed a predetermined threshold, then this assumption does not hold.

5. There is no COVID-19 data supporting the methodology.

6. Literature review is not sufficient. See Kim et al. (2009, Biometrics).

6. PLOS authors have the option to publish the peer review history of their article (what does this mean?). If published, this will include your full peer review and any attached files.

Reviewer #1: No

Reviewer #2: No

---

## [Author Response · Author response to Decision Letter 0]

13 Aug 2020

Dear Editor and Reviewer.

Thank you for you efforts and the points you raised, which we believe has lead to a significant improvement of our manuscript.

A detailed response to the points you raised are included in both the Cover letter and the Response Letter (which up to a heading are identical) 

We hope that the improvements of the manuscript now make it acceptable for publication.

Looking forward to hearing from you,

With kind regards

Rudolf Hanel

---

## [Editor Report · Decision Letter 1]

1 Oct 2020

Boosting test-efficiency by pooled testing for SARS-CoV-2 --

formula for optimal pool size

PONE-D-20-13830R1

Dear Dr. Hanel,

We’re pleased to inform you that your manuscript has been judged scientifically suitable for publication and will be formally accepted for publication once it meets all outstanding technical requirements.

Kind regards,

Oscar Millet

Academic Editor

PLOS ONE
---

## [Editor Report · Acceptance letter]

27 Oct 2020

PONE-D-20-13830R1 

Boosting test-efficiency by pooled testing for SARS-CoV-2 – formula for optimal pool size 

Dear Dr. Hanel:

I'm pleased to inform you that your manuscript has been deemed suitable for publication in PLOS ONE. Congratulations! Your manuscript is now with our production department. 

Kind regards, 

on behalf of

Dr. Oscar Millet 

Academic Editor

PLOS ONE